# Pregnancy-associated plasma protein-aa supports hair cell survival by regulating mitochondrial function

Mroj Alassaf[1,2], Emily C Daykin[1], Jaffna Mathiaparanam[1], Marc A Wolman[1]*

[1]Department of Integrative Biology, University of Wisconsin, Madison, United States; [2]Neuroscience Training Program, University of Wisconsin, Madison, United States

**Abstract** To support cell survival, mitochondria must balance energy production with oxidative stress. Inner ear hair cells are particularly vulnerable to oxidative stress; thus require tight mitochondrial regulation. We identified a novel molecular regulator of the hair cells' mitochondria and survival: Pregnancy-associated plasma protein-aa (Pappaa). Hair cells in zebrafish *pappaa* mutants exhibit mitochondrial defects, including elevated mitochondrial calcium, transmembrane potential, and reactive oxygen species (ROS) production and reduced antioxidant expression. In *pappaa* mutants, hair cell death is enhanced by stimulation of mitochondrial calcium or ROS production and suppressed by a mitochondrial ROS scavenger. As a secreted metalloprotease, Pappaa stimulates extracellular insulin-like growth factor 1 (IGF1) bioavailability. We found that the *pappaa* mutants' enhanced hair cell loss can be suppressed by stimulation of IGF1 availability and that Pappaa-IGF1 signaling acts post-developmentally to support hair cell survival. These results reveal Pappaa as an extracellular regulator of hair cell survival and essential mitochondrial function.
DOI: https://doi.org/10.7554/eLife.47061.001

*For correspondence: mawolman@wisc.edu

Competing interests: The authors declare that no competing interests exist.

## Introduction

Without a sufficient regenerative capacity, a nervous system's form and function critically depends on the molecular and cellular mechanisms that support its cells' longevity. Neural cell survival is inherently challenged by the nervous system's high energy demand, which is required to support basic functions, including maintaining membrane potential, propagating electrical signals, and coordinating the release and uptake of neurotransmitters (*Halliwell, 2006*; *Kann and Kovács, 2007*; *Howarth et al., 2012*). Metabolic energy is primarily supplied by mitochondrial oxidative phosphorylation (*Kann and Kovács, 2007*). Although this process is essential to cell survival, a cytotoxic consequence is the generation of reactive oxygen species (ROS). Oxidative stress caused by ROS accumulation damages vital cell components including DNA, proteins, and lipids (*Schieber and Chandel, 2014*). Neural cells are particularly vulnerable to oxidative stress due not only to their energy demand and thereby ROS production, but also to their relatively insufficient antioxidant capacity (*Halliwell, 1992*). This heightened susceptibility to oxidative stress-mediated cell death is believed to underlie aging and neurodegenerative disorders, including Alzheimer's disease (AD), Parkinson's disease (PD), and Amyotrophic lateral sclerosis (ALS) (*Perry et al., 2002*; *Barber et al., 2006*; *Mattson and Magnus, 2006*; *Blesa et al., 2015*).

Hair cells of the inner ear are a population of neural cells that are particularly susceptible to oxidative stress-induced death (*Gonzalez-Gonzalez, 2017*) These specialized sensory cells relay sound and balance information to the central nervous system. Hair cell death or damage, which is irreversible in mammals, is the primary cause of hearing loss, and is exacerbated by aging, genetic predisposition, exposure to loud noise and therapeutic agents (*Eggermont, 2017*). Identifying the

molecular and cellular mechanisms that promote the longevity of hair cells is a critical step towards designing therapeutic strategies that minimize the prevalence of hearing loss and its effect on quality of life. The insulin-like growth factor-1 (IGF1) signaling pathway is known to support mitochondrial function and cell survival (*García-Fernández et al., 2008*; *Lyons et al., 2017*). IGF1 deficiency has been shown to strongly correlate with age-related hearing loss in humans and animal models (*Riquelme et al., 2010*; *Lassale et al., 2017*). Recently, exogenous IGF1 supplementation was found to protect hair cells against death by exposure to the aminoglycoside neomycin (*Hayashi et al., 2013*; *Yamahara et al., 2017*). However, it remains unclear how endogenous IGF1 signaling is regulated to support hair cell survival and whether IGF1 signaling influences the hair cell's essential mitochondrial functions.

IGF1 is synthesized both in the liver for systemic distribution and locally in tissues, including the nervous system (*Bondy et al., 1992*; *Sjögren et al., 1999*). IGF1's biological functions are mediated by binding to cell surface IGF1 receptors (IGF1Rs), which act as receptor tyrosine kinases. When bound by IGF1, the IGF1R autophosphorylates and stimulates intracellular PI3kinase-Akt signaling (*Feldman et al., 1997*). Extracellularly, IGF1 is sequestered by IGF-binding proteins (IGFBPs), which restrict IGF1-IGF1R interactions (*Hwa et al., 1999*). To counter the inhibitory role of IGFBPs on IGF1 signaling, locally secreted proteases cleave IGFBPs to 'free' IGF1 and thereby stimulate local IGF1 signaling. One such protease, Pregnancy-associated plasma protein A (Pappa), targets a subset of IGFBPs to stimulate multiple IGF1-dependent processes, including synapse formation and function (*Boldt and Conover, 2007*; *Wolman et al., 2015*; *Miller et al., 2018*). Pappa has not been studied for its potential to act as an extracellular regulator of IGF1-dependent hair cell survival, mitochondrial function, or oxidative stress. Here, through analyses of lateral line hair cells in zebrafish *pappaa* mutants, we reveal a novel role for Pappaa in regulating mitochondrial function to support hair cell survival.

## Results

### IGF1R signaling affects hair cell survival and mitochondrial function in zebrafish

Hair cells of the zebrafish lateral line are found in superficial neuromasts and form a rosette-like structure that is surrounded by support cells (*Raible and Kruse, 2000*) (*Figure 1A*). These hair cells share functional, morphological, and molecular similarities with mammalian inner ear hair cells (*Ghysen and Dambly-Chaudière, 2007*). Acute exposure of larval zebrafish to the aminoglycoside neomycin triggers hair cell death and mitochondrial dysfunction (*Harris et al., 2003*; *Esterberg et al., 2014*; *Esterberg et al., 2016*). This experimental platform has been used to dissect the molecular and cellular mechanisms that support hair cell survival (*Owens et al., 2008*). A role for IGF1R signaling in the survival of zebrafish lateral line hair cells and their mitochondria has yet to be demonstrated. We hypothesized that if IGF1R signaling supports hair cell survival, then attenuating IGF1R signaling would further reduce hair cell survival following neomycin exposure. To test this, we used a transgenic line in which an inducible heat shock promoter drives ubiquitous expression of a dominant negative IGF1Ra [*Tg* (*hsp70:dnIGF1Ra-GFP*)] (*Kamei et al., 2011*). *dnIGF1Ra-GFP* expression was induced from 24 hr post fertilization (hpf) to 5 days post fertilization (dpf). At five dpf, larvae were exposed to neomycin for 1 hr and evaluated for hair cell survival 4 hr later. Larvae expressing *dnIGF1Ra-GFP* showed a greater reduction in hair cell survival compared to heat-shocked wild type and non-heat- shocked *Tg* (*hsp70:dnIGF1Ra-GFP*) larvae (*Figure 1B*).

Next, we evaluated mitochondrial activity in hair cells following attenuation of IGF1R signaling. A mitochondria's transmembrane potential is closely linked to its functions (*Zorova et al., 2018*) and can be visualized by the fluorescent, potentiometric probe TMRE. This live cationic dye readily accumulates in mitochondria based on the negative charge of their membrane potential. (*Crowley et al., 2016*). To determine whether IGF1R attenuation affected mitochondrial membrane potential, we treated wild type larvae with NVP-AEW541, a selective inhibitor of IGF1R phosphorylation (*Chablais and Jazwinska, 2010*), and loaded the hair cells with TMRE. We found that pharmacological attenuation of IGF1R signaling in wild type hair cells resulted in increased TMRE fluorescence (*Figure 1C–D'*). Together, these results indicate that IGF1R signaling regulates mitochondrial activity and the survival of zebrafish lateral line hair cells.

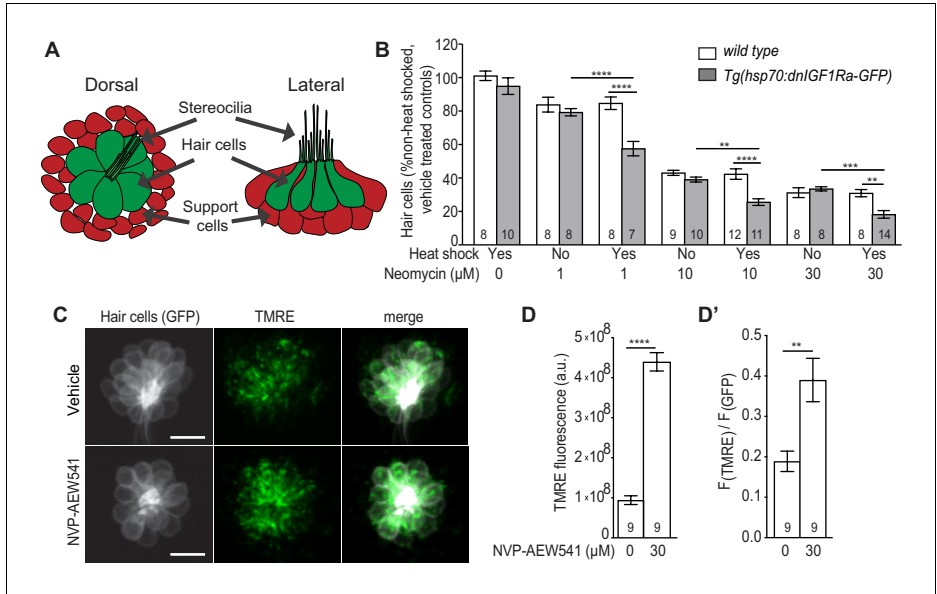

**Figure 1.** Inhibition of IGF1R activity enhances neomycin-induced hair cell death. (**A**) Schematic of lateral line neuromast. (**B**) Mean percentage of surviving hair cells following induction of *dnIGF1Ra-GFP* expression. To calculate hair cell survival percentage, hair cell number 4 hr post-neomycin treatment was normalized to mean hair cell number in non-heat-shocked, vehicle-treated larvae of the same genotype. **p<0.01, ***p<0.001, ****p<0.0001 two-way ANOVA, Holm-Sidak post test. N = 7–14 larvae per group (shown at base of bars), three neuromasts perlarva from two experiments. (**C**) *brn3c:GFP* labeled hair cells loaded with TMRE in NVP-AEW541 and vehicle treated larvae. (**D–D'**) Mean TMRE fluorescence (**D**) and mean TMRE fluorescence normalized to GFP fluorescence (**D'**) from Z-stack summation projections of *brn3c:GFP* labeled hair cells. N = 9 larvae per group. Total number of neuromasts included in the analysis = 26 (vehicle treated) and 27 (NVP-AEW541). **p<0.01, ****p<0.0001. Unpaired *t* test, Welch-corrected.

DOI: https://doi.org/10.7554/eLife.47061.002

The following source data is available for figure 1:

**Source data 1.** Hair cell survival post neomycin in wild type and *Tg(hsp70:dnIGF1Ra-GFP)* larvae.
DOI: https://doi.org/10.7554/eLife.47061.003

**Source data 2.** Mean F(TMRE) and ratio of mean F(TMRE) to mean F(GFP) in wild type and *pappaa^p170* hair cells following treatment with NVP-AEW541.
DOI: https://doi.org/10.7554/eLife.47061.004

## Pappaa is expressed by lateral line neuromast support cells

We were next curious whether extracellular regulation of IGF1 signaling was important for hair cell survival. A strong candidate to stimulate extracellular IGF1 availability is Pappaa (*Lawrence et al., 1999*). In situ hybridization revealed *pappaa* expression in lateral line neuromasts that co-localizes with the position of support cells, which surround the hair cell rosette (*Figure 2A–B*). To determine whether *pappaa* was also expressed by hair cells, but at levels below detection by fluorescent in situ hybridization (*Figure 2B*), we performed RT-PCR on fluorescently sorted hair cells from five dpf *Tg (brn3c:GFP)* (*Xiao et al., 2005*) larvae (*Figure 2C*). Again, we found that hair cells did not express *pappaa* (*Figure 2C*). Our in situ analysis also showed *pappaa* expression in the ventral spinal cord, where motor neurons reside (*Figure 2A*). As a control for fluorescent cell sorting and detection of *pappaa* by RT-PCR, we performed RT-PCR for *pappaa* on fluorescently sorted motor neurons from five dpf *Tg(mnx1:GFP)* larvae (*Rastegar et al., 2008*), and confirmed *pappaa* expression by motor neurons (*Figure 2C*).

## Pappaa supports hair cell survival

We next sought to determine whether Pappaa's regulation of IGF1 signaling supports hair cell survival. We examined hair cell survival after neomycin treatment of 5 dpf *pappaa* mutants (hereafter referred to as *pappaa^p170*). *pappaa^p170* mutants harbor nonsense mutations upstream of Pappaa's

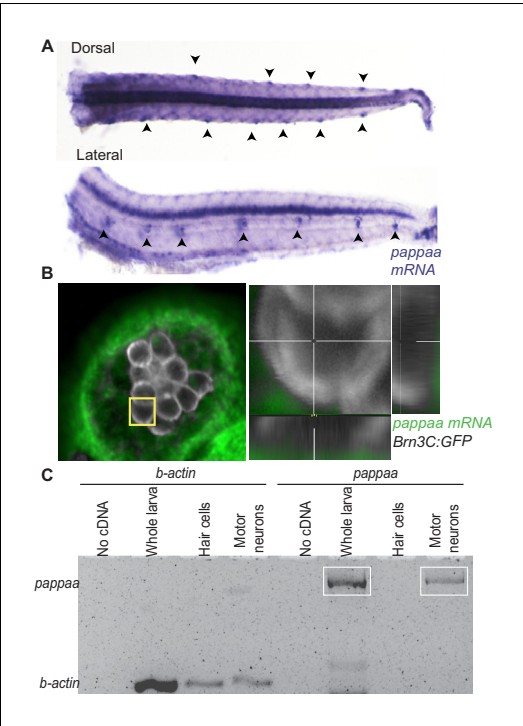

**Figure 2.** *pappaa* is expressed by neuromast support cells and motor neurons. (**A**) Whole mount in situ hybridization shows *pappaa* mRNA expression at four dpf by lateral line neuromasts (arrowheads). (**B**) Fluorescent in situ hybridization of *pappaa* (green) and *brn3c:GFP* labeled hair cells (white) shows *pappaa* mRNA expression by the support cells that surround hair cells. (**C**) RT-PCR of fluorescently sorted *brn3c:GFP* labeled hair cells and *mnx1:GFP* labeled motor neurons shows *pappaa* expression by motor neurons, but not hair cells. RT-PCR products represent *pappaa* cDNA fragment.

DOI: https://doi.org/10.7554/eLife.47061.005

proteolytic domain and show reduced IGF1R activation in other neural regions of *pappaa* expression (*Wolman et al., 2015*; *Miller et al., 2018*). Following exposure to neomycin, hair cells of *pappaa^p170* larvae showed reduced hair cell survival compared to wild type hair cells (*Figure 3A–B*). Notably, the support cells were unaffected by neomycin exposure in both genotypes (*Figure 3—figure supplement 1*). Next, we hypothesized that if Pappaa is acting through the IGF1 signaling pathway, then stimulating IGF1 signaling would improve hair cell survival in *pappaa^p170* larvae. To test this hypothesis, we bathed wild type and *pappaa^p170* larvae in recombinant human IGF1 protein. Pre-treatment with IGF1 for 24 hr prior to and during neomycin exposure improved hair cell survival in *pappaa^p170* larvae at concentrations of IGF1 that had no effect on hair cell survival in wild type larvae (*Figure 3C*). Because Pappaa acts to increase IGF1 bioavailability by freeing IGF1 from IGFBPs (*Boldt and Conover, 2007*), we asked whether this role was important for hair cell survival following neomycin exposure. To test this, we bathed wild type and *pappaa^p170* larvae in NBI-31772, an IGFBP inhibitor that stimulates IGF1 availability (*Safian et al., 2016*). Treatment with NBI-31772 for 24 hr prior to and during neomycin exposure improved hair cell survival in wild type and *pappaa^p170* larvae (*Figure 3D*). Taken together, these results suggest that extracellular regulation of IGF1 bioavailability by Pappaa enhances hair cell survival.

## Pappaa acts post-developmentally to promote hair cell survival

We next assessed when Pappaa acts to support hair cell survival. Zebrafish lateral line hair cells begin to appear at two dpf and are fully functional by four dpf (*Raible and Kruse, 2000*; *Ghysen and Dambly-Chaudière, 2007*). At five dpf, *pappaa^p170* larvae are responsive to acoustic stimuli (*Wolman et al., 2015*), suggesting that their hair cells are functionally intact. In addition, *pappaa^p170* hair cells appeared morphologically indistinguishable from hair cells in wild type larvae (*Figure 4A*). Therefore, we hypothesized that Pappaa acts post-developmentally to support hair cell survival. To test this idea, we asked whether post-developmental expression of Pappaa was sufficient to suppress the *pappaa* mutant's enhanced hair cell loss when exposed to neomycin. We generated a transgenic line in which a temporally inducible heat shock promoter drives ubiquitous expression of Pappaa (*Tg(hsp70:pappaa-GFP)*). We found that induced expression of Pappaa, beginning at four dpf and through neomycin treatment at five dpf, resulted in the complete rescue of *pappaa^p170* hair cell sensitivity to neomycin and raised *pappaa^p170* hair cell survival to wild type levels (*Figure 4B*). Consistent with these results, we found that post-developmental attenuation of IGF1R signaling, through induction of *dnIGF1Ra-GFP* expression beginning at four dpf, was sufficient to reduce hair cell survival when exposed to neomycin at five dpf (*Figure 4C*).

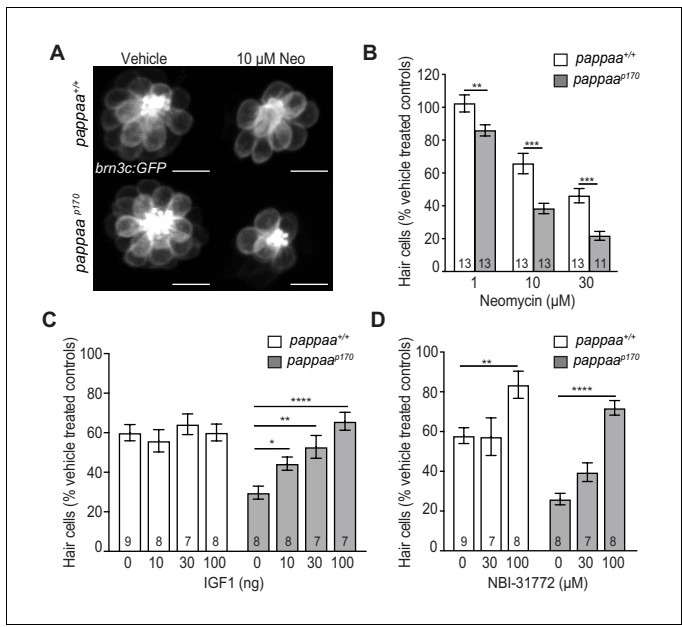

**Figure 3.** Hair cell survival is reduced in zebrafish *pappaa^{p170}* larvae. (**A**) Representative images of *brn3c:GFP* labeled hair cells from vehicle or 10 μM neomycin treated larvae. Scale = 10 μm. (**B**) Mean percentage of surviving hair cells. To calculate hair cell survival percentage, hair cell number 4 hr post-neomycin treatment was normalized to mean hair cell number in vehicle treated larvae of the same genotype. **p<0.01, ***p<0.001, two-way ANOVA, Holm-Sidak post test. N = 11–15 larvae per group (shown at base of bars). Total number of neuromasts included in the analysis = 45 (wild type; vehicle-treated), 39 (wild type; 1 μM neomycin), 39 (wild type; 10 μM neomycin), 39 (wild type; 30 μM neomycin), 45 (*pappaa^{p170}*; vehicle-treated), 39 (*pappaa^{p170}*; 1 μM neomycin), 39 (*pappaa^{p170}*; 10 μM neomycin), and 33 (*pappaa^{p170}*; 30 μM neomycin) from two experiments. (**C**) Mean percentage of surviving hair cells following co-treatment with IGF1 and 10 μM neomycin. To calculate hair cell survival percentage, hair cell counts after treatment were normalized to hair cell number in vehicle treated larvae of same genotype. *<0.05, **p<0.01 ****p<0.0001. Two-way ANOVA, Holm-Sidak post test. N = 7–11 larvae per group (shown at base of bars). Total number of neuromasts included in the analysis = 33 (wild type; vehicle-treated), 27 (wild type; 10 μM neomycin), 24 (wild type; 10 μM neomycin +10 ng IGF1), 21 (wild type; 10 μM neomycin +30 ng IGF1), 24 (wild type; 10 μM neomycin +100 ng IGF1), 33 (*pappaa^{p170}*; vehicle-treated), 24 (*pappaa^{p170}*; 10 μM neomycin), 24 (*pappaa^{p170}*; 10 μM neomycin +10 ng IGF1), 21 (*pappaa^{p170}*; 10 μM neomycin +30 ng IGF1), and 21 (*pappaa^{p170}*; 10 μM neomycin +100 ng IGF1) (**D**) Mean percentage of surviving hair cells following co-treatment with NBI-31772 and 10 μM neomycin. To calculate hair cell survival percentage, hair cell counts after treatment were normalized to hair cell number in vehicle treated larvae of same genotype. **p<0.01 ****p<0.0001. Two-way ANOVA, Holm-Sidak post test. N = 7–9 larvae per group (shown at base of bars). Total number of neuromasts included in the analysis = 15 (wild type; vehicle-treated), 27 (wild type; 10 μM neomycin), 21 (wild type; 10 μM neomycin +30 μM NBI-31772), 24 (wild type; 10 μM neomycin +100 μM NBI-31772), 15 (*pappaa^{p170}*; vehicle-treated), 24 (*pappaa^{p170}*; 10 μM neomycin), 21 (*pappaa^{p170}*; 10 μM neomycin +30 μM NBI-31772), and 24 (*pappaa^{p170}*; 10 μM neomycin +100 μM NBI-31772).

DOI: https://doi.org/10.7554/eLife.47061.006

The following source data and figure supplements are available for figure 3:

**Source data 1.** Hair cell survival post neomycin in wild type and *pappaa^{p170}* larvae.
DOI: https://doi.org/10.7554/eLife.47061.009

**Source data 2.** Hair cell survival post co-treatment of IGF1 and neomycin in wild type and *pappaa^{p170}* larvae.
DOI: https://doi.org/10.7554/eLife.47061.010

**Source data 3.** Hair cell survival post co-treatment of NBI-31772 and neomycin in wild type and *pappaa^{p170}* larvae.
DOI: https://doi.org/10.7554/eLife.47061.011

**Figure supplement 1.** Support cells are not affected by neomycin treatment.
DOI: https://doi.org/10.7554/eLife.47061.007

**Figure supplement 1—source data 1.** Support cell survival post neomycin in wild type and *pappaa^{p170}*larvae.
DOI: https://doi.org/10.7554/eLife.47061.008

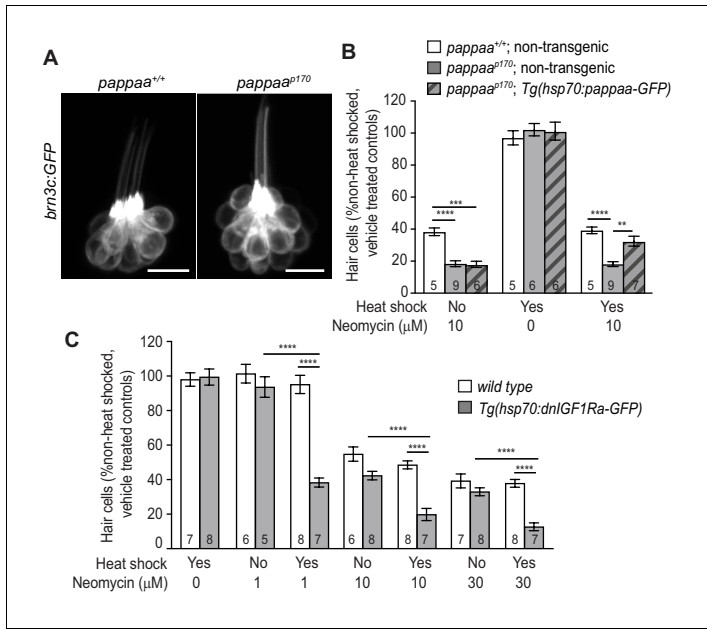

**Figure 4.** Post-developmental regulation of IGF1 signaling by Pappaa is required for hair cell survival. (**A**) Lateral view of *brn3c:GFP* labeled hair cells in 5dpf wild type and *pappaa^p170* larvae. Scale = 10 µm. (**B**) Mean percentage of surviving hair cells following post-developmental induction of Pappaa expression in *pappaa^p170* larvae. To calculate hair cell survival percentage, hair cell number 4 hr post-neomycin treatment was normalized to mean hair cell number in non-heat-shocked, vehicle-treated larvae of the same genotype. **p<0.01, ***p<0.001, ****p<0.0001, 2-way ANOVA, Holm-Sidak post test. N = 5–9 larvae per group (shown at base of bars), three neuromasts perlarva. (**C**) Mean percentage of surviving hair cells following post-developmental induction of *dnIGF1Ra-GFP* expression. To calculate hair cell survival percentage, hair cell number 4 hr post-neomycin treatment was normalized to mean hair cell number in non-heat-shocked, vehicle-treated larvae of the same genotype. ****p<0.0001, two-way ANOVA, Holm-Sidak post test. N = number of larvae per group (shown at base of bars), three neuromasts per larva.

DOI: https://doi.org/10.7554/eLife.47061.012

The following source data is available for figure 4:

**Source data 1.** Hair cell survival post neomycin in wild type, *pappaa^p170*, and *Tg(hsp70:pappaa-GFP); pappaa^p170* larvae.

DOI: https://doi.org/10.7554/eLife.47061.013

**Source data 2.** Hair cell survival post neomycin in wild type and *Tg(hsp70:dnIGF1Ra-GFP)* larvae.

DOI: https://doi.org/10.7554/eLife.47061.014

## Pappaa loss causes increased mitochondrial ROS in hair cells

A role for Pappaa in hair cell survival is novel. To define how Pappaa activity influences hair cell survival, we evaluated cellular mechanisms known to underlie their neomycin-induced death. Neomycin enters hair cells via mechanotransduction (MET) channels found on the tips of stereocilia (*Kroese et al., 1989*). MET channel permeability has been correlated to hair cells' neomycin sensitivity (*Alharazneh et al., 2011*; *Stawicki et al., 2016*). We hypothesized that *pappaa^p170* hair cells may be more susceptible to neomycin-induced death due to an increase in MET channel-mediated entry. To assess MET channel entry we compared uptake of FM1-43, a fluorescent styryl dye that enters cells through MET channels (*Meyers et al., 2003*), by hair cells in wild type and *pappaa^p170* larvae. FM1-43 fluorescence was equivalent between wild type and *pappaa^p170* hair cells (*Figure 5A–B'*), suggesting that the increased death of *pappaa^p170* hair cells was not due to increased MET channel permeability.

We next questioned whether Pappaa affects essential organelle functions in hair cells, which are known to be disrupted by neomycin. Within the hair cell, neomycin triggers $Ca^{2+}$ transfer from the endoplasmic reticulum to the mitochondria (*Esterberg et al., 2014*). This $Ca^{2+}$ transfer results in stimulation of the mitochondrial respiratory chain, increased mitochondrial transmembrane potential,

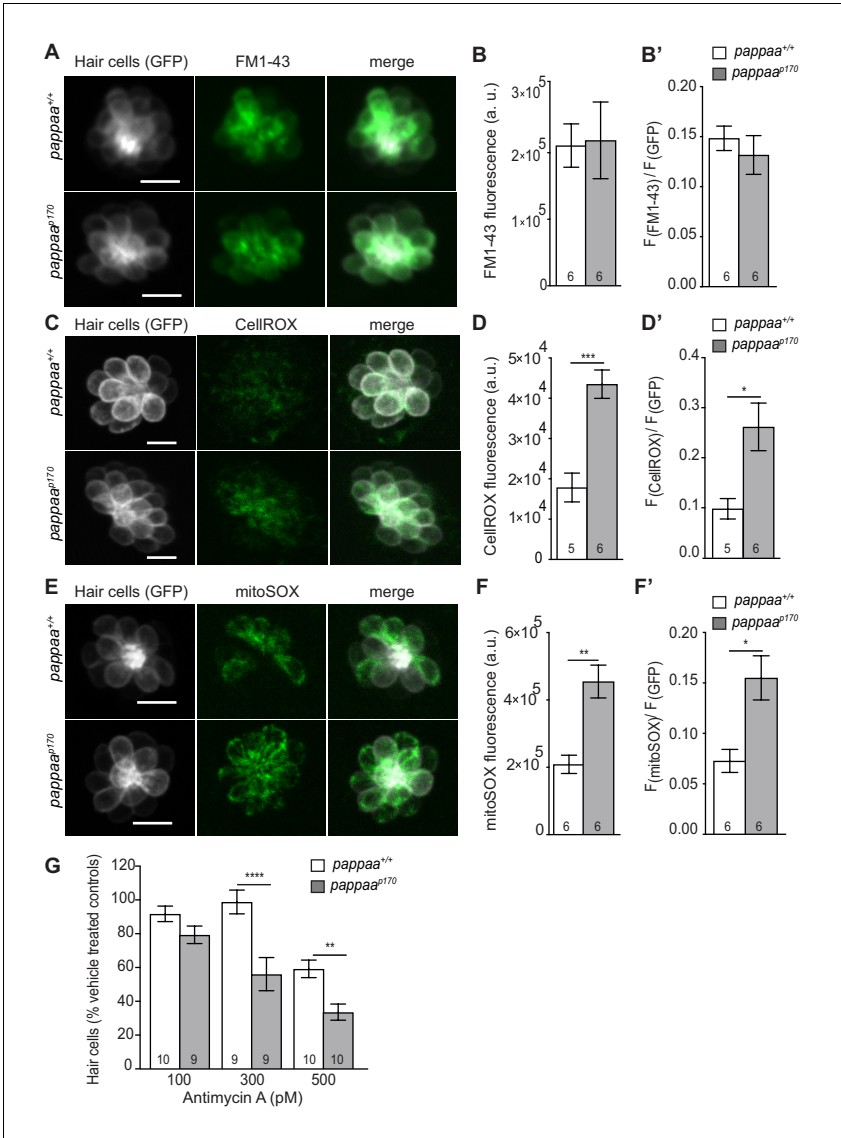

**Figure 5.** Pappaa regulates mitochondrial ROS generation. (**A, C, E**) Still images of live *brn3c:GFP* hair cells loaded with the amphypathic styryl dye FM1-43 (**A**) or cytoplasmic or mitochondrial ROS indicators (C: CellROX, E: mitoSOX). Scale = 10 μm. (**B, D, F**) Mean dye fluorescence (**D**) and mean dye fluorescence normalized to GFP fluorescence (**B', D', F'**) from Z-stack summation projections of *brn3c:GFP* labeled hair cells. N = 5–6 larvae per group (shown at base of bars). Total number of neuromasts included in the analysis = 18 (wild type; FM1-43), 18 (*pappaa^p170*; FM1-43), 21 (wild type; CellROX), 21 (*pappaa^p170*; CellROX), 18 (wild type; mitoSOX), and 22 (*pappaa^p170*; mitoSOX). *p<0.05, **<*p* < 0.01, ***p<0.001. Unpaired *t* test, Welch-corrected. (**G**) Mean percentage of surviving hair cells post Antimycin A treatment. To calculate hair cell survival percentage, hair cell counts after treatment were normalized to hair cell number in vehicle treated larvae of same genotype. **p<0.01 ****p<0.0001. Two-way ANOVA, Holm-Sidak post test. N = 9–10 larvae per group (shown at base of bars). Total number of neuromasts included in the analysis = 30 (wild type; vehicle-treated), 30 (wild type; 100pM antimycin a), 27 (wild type; 300pM antimycin a), 30 (wild type; 500pM antimycin a), 27 (*pappaa^p170*; vehicle-treated), 27 (*pappaa^p170*; 100pM antimycin a), 27 (*pappaa^p170*; 300pM antimycin a), and 30 (*pappaa^p170*; 500pM antimycin a).

DOI: https://doi.org/10.7554/eLife.47061.015

The following source data and figure supplements are available for figure 5:

**Source data 1.** Mean F(FM1-43) and ratio of mean F(FM1-43) to F(GFP) in wild type and *pappaa^p170* hair cells.
DOI: https://doi.org/10.7554/eLife.47061.019
**Source data 2.** Mean F(CellROX) and ratio of mean F(CellROX) to F(GFP) in wild type and *pappaa^p170* hair cells.
DOI: https://doi.org/10.7554/eLife.47061.020
*Figure 5 continued on next page*

*Figure 5 continued*

**Source data 3.** Mean F(mitoSOX) and ratio of mean F(mitoSOX) to F(GFP) in wild type and *pappaa^p170* hair cells.
DOI: https://doi.org/10.7554/eLife.47061.021
**Source data 4.** Hair cell survival post Antimycin A in wild type and *pappaa^p170* larvae.
DOI: https://doi.org/10.7554/eLife.47061.022
**Figure supplement 1.** Mitochondrial mass is equivalent in wild type and *pappaa^p170* hair cells.
DOI: https://doi.org/10.7554/eLife.47061.016
**Figure supplement 1—source data 1.** Mean F(mitotracker) in wild type and *pappaa^p170* hair cells.
DOI: https://doi.org/10.7554/eLife.47061.017
**Figure supplement 1—source data 2.** Mean F(CellROX) and ratio of mean F(CellROX) to mean F(GFP) in wild type and pappaa mutant hair cells.
DOI: https://doi.org/10.7554/eLife.47061.018

and increased ROS production (*Görlach et al., 2015*; *Esterberg et al., 2016*). The ensuing oxidative stress ultimately underlies the neomycin's cytotoxic effect on hair cells. To explore whether excessive ROS production underlies *pappaa^p170* hair cells' increased sensitivity to neomycin, we evaluated cytoplasmic ROS levels with a live fluorescent indicator of ROS (CellROX) (*Esterberg et al., 2016*). *pappaa^p170* hair cells displayed elevated ROS levels at baseline; prior to addition of neomycin (*Figure 5C–D′*). Given that the mitochondria are the primary generators of cellular ROS (*Lenaz, 2001*), we asked whether the elevated levels of cytoplasmic ROS observed in *pappaa^p170* hair cells originated from the mitochondria. We evaluated mitochondrial ROS with the live fluorescent indicator mitoSOX (*Esterberg et al., 2016*), again without neomycin treatment, and observed increased signal in hair cells of *pappaa^p170* compared to wild type (*Figure 5E–F′*). This increased mitochondrial ROS was not due to an overabundance of mitochondria within *pappaa^p170* hair cells, as determined by measuring mitochondrial mass with mitotracker (*Figure 5—figure supplement 1*).

We hypothesized that the elevated ROS in *pappaa^p170* hair cells predisposed them closer to a cytotoxic threshold of oxidative stress. To test this idea, we asked whether *pappaa^p170* showed reduced hair cell survival following pharmacological stimulation of mitochondrial ROS via exposure to Antimycin A, an inhibitor of the mitochondrial electron transport chain (*Hoegger et al., 2008*; *Quinlan et al., 2011*) We found that *pappaa^p170* hair cells showed increased death by Antimycin A compared to wild type hair cells (*Figure 5G*). These results are consistent with the idea that *pappaa^p170* hair cells are predisposed to oxidative stress-induced death due to elevated baseline levels of ROS.

## Pappaa regulates mitochondrial Ca$^{2+}$ uptake and transmembrane potential

Mitochondrial ROS production is stimulated by Ca$^{2+}$ entry into the mitochondria (*Brookes et al., 2004*; *Görlach et al., 2015*). Given the increased mitochondrial ROS in *pappaa^p170* hair cells, we asked whether the mutants' hair cell mitochondria exhibited increased Ca$^{2+}$ levels. To address this, we used a transgenic line *Tg(myo6b:mitoGCaMP3)*, in which a mitochondria-targeted genetically encoded Ca$^{2+}$ indicator (*GCaMP3*) is expressed in hair cells (*Esterberg et al., 2014*). Live imaging of mitoGCaMP3 fluorescence revealed a 2-fold increase in fluorescent intensity in *pappaa^p170* hair cells compared to wild type hair cells (*Figure 6A–B*). Mitochondrial Ca$^{2+}$ uptake is driven by the negative electrochemical gradient of the mitochondrial transmembrane potential, a product of mitochondrial respiration. Ca$^{2+}$-induced stimulation of mitochondrial oxidative phosphorylation causes further hyperpolarization of mitochondrial transmembrane potential, leading to increased uptake of Ca$^{2+}$ (*Brookes et al., 2004*; *Adam-Vizi and Starkov, 2010*; *Ivannikov and Macleod, 2013*; *Esterberg et al., 2014*; *Görlach et al., 2015*). Therefore, we hypothesized that *pappaa^p170* mitochondria would have a more negative transmembrane potential compared to wild type. Using TMRE as an indicator of mitochondrial transmembrane potential (*Perry et al., 2011*), we found that *pappaa^p170* mitochondria possess a more negative transmembrane potential compared to wild type (*Figure 6C–D′*). This increased TMRE signal is similar to our observations following pharmacological inhibition of IGF1R (*Figure 1C–D′*).

Given that mitochondria of *pappaa^p170* hair cells exhibited elevated Ca$^{2+}$ (*Figure 6A–B*) and a more negative transmembrane potential (*Figure 6C–D′*), we hypothesized that pharmacologically

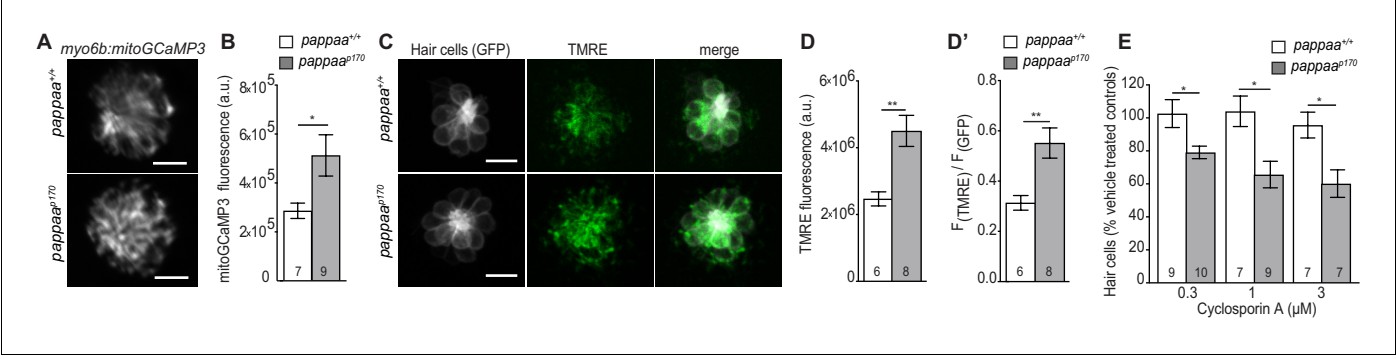

**Figure 6.** Mitochondrial Ca$^{2+}$ levels and transmembrane potential are disrupted in *pappaa*$^{p170}$ hair cells. (**A**) Still images from live *myo6b:mitoGCaMP3* labeled hair cells. Scale = 10 μm. (**B**) Mean mitoGCaMP fluorescence; quantified from Z-stack summation projection. *p<0.05. Unpaired *t* test, Welch-corrected. N = 7–9 larvae per group (shown at base of bars). Total number of neuromasts included in the analysis = 19 (wild type) and 26 (*pappaa*$^{p170}$). (**C**) Still images from live *brn3c:GFP* labeled hair cells loaded with TMRE. Scale = 10 μm. (**D**) Mean TMRE fluorescence (**D**) and mean TMRE fluorescence normalized to GFP fluorescence (**D'**) from Z-stack summation projections of *brn3c:GFP* labeled hair cells.. N = 6–8 larvae per group (shown at base of bars). Total number of neuromasts included in the analysis = 17 (wild type) and 23 (*pappaa*$^{p170}$).. **p<0.01. Unpaired *t* test, Welch-corrected. (**E**) Mean percentage of surviving hair cells post Cyclosporin A treatment. To calculate hair cell survival percentage, hair cell counts post-treatment were normalized to hair cell numbers in vehicle treated larvae of same genotype. N = 7–10 larvae per group (shown at base of bars). Total number of neuromasts included in the analysis = 45 (wild type; vehicle-treated), 27 (wild type; 0.1 μM CsA), 21 (wild type; 1 μM CsA), 21 (wild type; 3 μM CsA), 42 (*pappaa*$^{p170}$; vehicle-treated), 30 (*pappaa*$^{p170}$; 0.1 μM CsA), 27 (*pappaa*$^{p170}$; 1 μM CsA), and 21 (*pappaa*$^{p170}$; 3 μM CsA) from two experiments. *p<0.05. Two-way ANOVA, Holm-Sidak post test.

DOI: https://doi.org/10.7554/eLife.47061.023

The following source data is available for figure 6:

**Source data 1.** Mean F(mitoGCaMP) in wild type and *pappaa*$^{p170}$ hair cells.
DOI: https://doi.org/10.7554/eLife.47061.024
**Source data 2.** Mean F(TMRE) and ratio of mean F(TMRE) to F(GFP) in wild type and *pappaa*$^{p170}$ hair cells.
DOI: https://doi.org/10.7554/eLife.47061.025
**Source data 3.** Hair cell survival post Cyclosporin A in wild type and *pappaa*$^{p170}$ larvae.
DOI: https://doi.org/10.7554/eLife.47061.026

disrupting these mitochondrial features would have a more cytotoxic effect on *pappaa*$^{p170}$ hair cells. To test this idea, we exposed wild type and *pappaa*$^{p170}$ larvae to Cyclosporin A (CsA), an inhibitor of the mitochondrial permeability transition pore that causes buildup of mitochondrial Ca$^{2+}$ and further hyperpolarizes mitochondria (*Crompton et al., 1988*; *Esterberg et al., 2014*). *pappaa*$^{p170}$ larvae showed reduced hair cell survival at concentrations of CsA, which had no effect on hair cell survival in wild type larvae (*Figure 6E*). Taken together, these results suggest that Pappaa loss disrupts mitochondrial functions that can predispose hair cells to death.

## Pappaa regulates the expression of mitochondrial antioxidants

Oxidative stress can be caused by an imbalance in ROS production and antioxidant activity (*Betteridge, 2000*). IGF1 signaling positively correlates with antioxidant expression (*Higashi et al., 2013*; *Wang et al., 2016*). Therefore, we questioned whether the cytotoxicity of excessive ROS in *pappaa*$^{p170}$ was compounded by a reduced antioxidant system. To address this, we compared gene expression of antioxidants in wild type and *pappaa*$^{p170}$ hair cells by RT-qPCR. This analysis revealed reduced expression of mitochondrial antioxidants genes (*gpx*, *sod1*, and *sod2*) (*Figure 7A*) (*Weisiger and Fridovich, 1973*; *Okado-Matsumoto and Fridovich, 2001*; *Higgins et al., 2002*; *Brigelius-Flohé and Maiorino, 2013*), but not expression of *catalase*; an antioxidant that does not localize to mitochondria (*Zhou and Kang, 2000*). These results suggest that the *pappaa*$^{p170}$ hair cells' elevated ROS can be attributed not only to increased mitochondrial calcium and transmembrane potential, but also to reduced mitochondrial antioxidants. Finally, we asked whether the increased mitochondria-generated ROS in *pappaa*$^{p170}$ hair cells was sufficient to explain their increased mortality rate when exposed to neomycin. We hypothesized that if this were the case, then reducing mitochondrial-ROS would suppress their increased mortality. To test this idea we

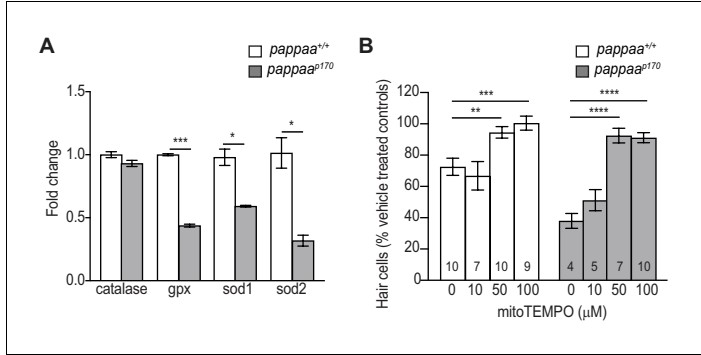

**Figure 7.** Pappaa regulates expression of mitochondrial antioxidants. (**A**) Mean fold change in antioxidants transcript levels in hair cells at five dpf. cDNA was from FACsorted hair cells that were collected from 200 *Tg (brn3c:GFP)* dissected tails. N = 2–4 technical replicates/gene. *p<0.05, ***p<0.001. Multiple *t* tests, Holm-Sidak post test. Error bars = SEM. (**B**) Mean percentage of surviving *pappaa^p170* hair cells following co-treatment with mitoTEMPO and 10 µM neomycin. To calculate hair cell survival percentage, hair cell counts 4 hr post-neomycin treatment were normalized to hair cell counts in vehicle treated *pappaa^p170* larvae. **p<0.01, ***p<0.001, ****p<0.0001. Two-way ANOVA, Holm-Sidak post test. N = 4–10 (shown at base of bars). Total number of neuromasts included in the analysis = 30 (wild type; vehicle-treated), 30 (wild type; 10 µM neomycin), 21 (wild type; 10 µM neomycin +10 µM mitoTEMPO), 30 (wild type; 10 µM neomycin +50 µM mitoTEMPO), 27 (wild type; 10 µM neomycin +100 µM mitoTEMPO), 30 (*pappaa^p170*; vehicle-treated), 12 (*pappaa^p170*; 10 µM neomycin), 15 (*pappaa^p170*; 10 µM neomycin +10 µM mitoTEMPO), 21 (*pappaa^p170*; 10 µM neomycin +50 µM mitoTEMPO), and 30 (*pappaa^p170*; 100 µM neomycin +10 µM mitoTEMPO). Error bars = SEM.

DOI: https://doi.org/10.7554/eLife.47061.027

The following source data is available for figure 7:

**Source data 1.** Quantification of antioxidant transcript expression in wild type and *pappaa^p170* hair cells.
DOI: https://doi.org/10.7554/eLife.47061.028

**Source data 2.** Hair cell survival post co-treatment of mitoTEMPO and neomycin in *pappaa^p170* larvae.
DOI: https://doi.org/10.7554/eLife.47061.029

exposed *pappaa^p170* larvae to the mitochondria-targeted ROS scavenger mitoTEMPO (*Esterberg et al., 2016*) and observed up to complete protection of *pappaa^p170* hair cells against neomycin-induced death (*Figure 7B*). These results suggest that abnormally elevated mitochondrial ROS underlies the enhanced hair cell death in neomycin treated *pappaa^p170* zebrafish.

## Discussion

Extracellular factors and the regulation of mitochondrial function and oxidative stress have been demonstrated to support the survival of various cell types (*Hasan et al., 2003*; *Echave et al., 2009*; *Li et al., 2009*; *Wang et al., 2013*; *Genis et al., 2014*; *Pyakurel et al., 2015*; *Kim, 2017*). For cells that do not have a capacity for regeneration, like hair cells, the regulation of these factors and intracellular processes are particularly important. Exogenous application of IGF1 was recently shown to protect hair cells against neomycin exposure (*Hayashi et al., 2013*). This finding identified a molecular pathway that could be potentially targeted to combat sensorineural hearing loss. Yet, significant questions remained regarding the mechanism by which IGF1 signaling serves this role. For example, it was unclear whether endogenous IGF1 signaling supports hair cell survival and how this pathway is extracellularly regulated to promote hair cell survival. Through zebrafish mutant analysis, we identified a novel extracellular regulator of IGF1 signaling that supports hair cell survival and mitochondrial function: the secreted metalloprotease Pappaa. Based on a series of in vivo experiments we propose a model by which Pappaa stimulates IGF1R signaling in hair cells to control mitochondrial function and oxidative stress, and thereby, promotes the longevity of these cells.

### Pappaa regulated IGF1 signaling supports hair cell survival

Pappaa acts as an extracellular positive regulator of IGF1 signaling by cleaving inhibitory IGFBPs, thereby freeing IGF1 to bind and activate cell-surface IGF1Rs (*Boldt and Conover, 2007*). In the

nervous system, Pappaa's signaling role has been shown to support synaptic structure and function, but a cell protective function had not been explored. Our results demonstrate that *pappaa^p170* hair cells showed increased mortality to neomycin (*Figure 3A–B*), and that this phenotype could be suppressed by pharmacological stimulation of IGF1 availability (*Figure 3C and D*). Given the novelty for Pappaa in supporting hair cell survival, it is interesting to consider whether Pappaa acts developmentally or post-developmentally in this context. In five dpf *pappaa^p170*, hair cells appeared to develop normally based on their cellular morphology (*Figure 4A*) and ability to mediate acoustic startle responses (*Wolman et al., 2015*). We found that post-developmental expression of Pappaa was sufficient to increase the *pappaa^p170* hair cells' survival when exposed to neomycin to near wild type levels (*Figure 4B*). Consistent with this post-developmental role for Pappaa, post-developmental attenuation of IGF1R signaling was also sufficient to increase neomycin-induced hair cell loss (*Figure 4C*), while stimulation of IGF1R signaling was sufficient to suppress *pappaa^p170* hair cell loss (*Figure 3C and D*). Taken together, these findings suggest that Pappaa-IGF1R signaling acts post-developmentally to mediate resistance against toxins, like neomycin.

To support hair cell survival, *pappaa's* expression pattern suggests that Pappaa is likely to act in a paracrine manner. Although hair cells require Pappaa for survival, they do not express *pappaa*. Rather, *pappaa* is expressed by the adjacent support cells (*Figure 2A–C*). Support cells have been shown to secrete factors that promote hair cell survival (*May et al., 2013*; *Yamahara et al., 2017*) and our results suggest that Pappaa is one such factor. To understand Pappaa's cell autonomy it will be necessary to define in which cells IGF1 signaling activation is required. It is possible that Pappaa does not act directly on hair cells, rather it may influence support cells to promote hair cell survival. Moreover, it will be interesting to define the molecular cues that trigger Pappaa activity, their cellular source, and to determine whether Pappaa acts directly in response to such cues or serves a more preventative role for hair cells.

## Pappaa affects mitochondrial function and oxidative stress

To understand how a Pappaa-IGF1 signaling deficiency increased neomycin-induced hair cell loss, we examined the hair cells' mitochondria, which are known to be disrupted by neomycin (*Esterberg et al., 2016*). The mitochondria in *pappaa^p170* hair cells showed multiple signs of dysfunction, including elevated ROS (*Figure 5E–F'*), transmembrane potential (*Figure 6C–D'*), and $Ca^{2+}$ load (*Figure 6A–B*). Consistent with these observations, reduced IGF1 signaling has been associated with increased ROS production and oxidative stress (*García-Fernández et al., 2008*; *Lyons et al., 2017*). Three lines of evidence suggest that mitochondrial dysfunction, and particularly the elevated ROS levels, underlie the increased hair cell loss in *pappaa^p170*. First, *pappaa^p170* hair cells showed enhanced sensitivity to pharmacological stimulators of mitochondrial ROS production (*Figures 5G and 6E*). Second, *pappaa^p170* hair cells showed reduced expression of mitochondrial antioxidant genes (*Figure 7A*). Third, attenuation of mitochondrial ROS was sufficient to suppress neomycin-induced hair cell loss in *pappaa^p170* (*Figure 7B*).

Based on results presented here, we can only speculate on the primary subcellular locus and defect that triggers mitochondrial dysfunction in *pappaa^p170* hair cells. The challenge lies in the tight interplay between mitochondrial transmembrane potential, $Ca^{2+}$ load, and ROS production and clearance (*Brookes et al., 2004*; *Adam-Vizi and Starkov, 2010*; *Ivannikov and Macleod, 2013*; *Esterberg et al., 2014*; *Görlach et al., 2015*). The oxidative phosphorylation that generates ROS relies on maintaining a negative mitochondrial transmembrane potential. Negative transmembrane potential is achieved by pumping protons out of the mitochondrial matrix as electrons move across the electron transport chain. Protons then move down the electrochemical gradient through ATP synthase to produce ATP. Given that ROS is a byproduct of oxidative phosphorylation, a more negative transmembrane potential yields more ROS (*Kann and Kovács, 2007*; *Zorov et al., 2014*). Mitochondrial $Ca^{2+}$ is a key regulator of transmembrane potential and the resultant ROS generation, as it stimulates the activity of key enzymes involved in oxidative phosphorylation (*Brookes et al., 2004*). And, $Ca^{2+}$ uptake by the mitochondria is driven by the electrochemical gradient of a negative transmembrane potential. Thus, $Ca^{2+}$ and transmembrane potential are locked in a feedback loop (*Brookes et al., 2004*; *Adam-Vizi and Starkov, 2010*; *Ivannikov and Macleod, 2013*; *Esterberg et al., 2014*; *Görlach et al., 2015*). Because mitochondria in *pappaa^p170* hair cells have a more negative transmembrane potential (*Figure 6C–D'*) and experience $Ca^{2+}$ overload (*Figure 6A–B*), this likely sensitizes the mitochondria to any further increase in $Ca^{2+}$ levels. In support of this

idea, *pappaa^p170* hair cells were hypersensitive to Cyclosporin A (**Figure 6E**), which increases mitochondrial $Ca^{2+}$ levels by blocking the mitochondrial permeability transition pore (**Smaili and Russell, 1999**).

Given that oxidative stress is caused by the imbalance between ROS production and clearance (**Betteridge, 2000**), *pappaa^p170* mitochondrial dysfunction may have been triggered by their weak antioxidant system (**Figure 7A**). As the main site for ROS generation, mitochondria are primary targets of ROS-induced damage. Specifically, ROS can damage the mitochondrial oxidative phosphorylation machinery leading to excessive electron 'leaks' and the ensuing ROS formation (**Marchi et al., 2012**). To prevent such oxidative damage, mitochondria are equipped with antioxidants that can rapidly neutralize ROS by converting them into water molecules. Failure of antioxidants to clear ROS, either due to reduced enzymatic activity or reduced expression levels, has been shown to damage several components of the mitochondria causing oxidative stress that culminates in cell death (**Williams et al., 1998**; **Armstrong and Jones, 2002**; **Aquilano et al., 2006**; **Velarde et al., 2012**). Therefore, it is no surprise that mitochondria-targeted antioxidants have shown great promise in clinical studies to treat neurodegeneration (**Sheu et al., 2006**). Indeed, mitochondria-targeted antioxidant treatment was successful in preventing ROS-induced hair cell death (**Figure 7B**). Given that ROS can modulate the activity of $Ca^{2+}$ channels and consequently raise mitochondrial $Ca^{2+}$ load and transmembrane potential (**Chaube and Werstuck, 2016**), it is possible that the reduced mitochondrial antioxidants levels in *pappaa^p170* hair cells triggered their mitochondrial dysfunction resulting in a vicious cycle of further ROS production. Alternatively, *pappaa^p170* mitochondria may suffer from overproduction of ROS that is compounded by insufficient clearance. Further experimental dissections of mitochondria in *pappaa^p17* mutant hair cells are needed to define the primary locus by which a deficiency in Pappaa-IGF1 signaling alters subcellular processes in hair cells.

## Conclusions and outlook

Here, we define a novel role for Pappaa in hair cell survival. Although ample evidence exists about the protective role of IGF1, little is known about how IGF1's extracellular availability is regulated to promote cell survival. Our results demonstrate that extracellular regulation of IGF1 by Pappaa is critical for maintaining mitochondrial function, and in turn, survival of hair cells. Future work should explore whether Pappaa plays a similar role in other neural cell types, including motor neurons and the retinal cells (**Miller et al., 2018**), where IGF1 signaling is known to affect cell survival (**Lewis et al., 1993**; **Sakowski et al., 2009**) Furthermore, future experimentation will be needed to resolve the cellular autonomy of Pappaa-IGF1R signaling to cell survival and to define the primary subcellular locus by which this signaling axis influences mitochondrial activity and oxidative stress.

# Materials and methods

**Key resources table**

| Reagent type (species) or resource | Designation | Source or reference | Identifiers | Additional information |
|---|---|---|---|---|
| Gene (*Danio rerio*) | *pappaa^p170* | **Wolman et al., 2015** | RRID:ZFIN_ZDB-GENO-190322-4 | single nucleotide nonsense mutation C > T at position 964 in Exon 3 |
| Strain, strain background (*Danio rerio*) | Tg(myo6b:mitoGCaMP3) | **Esterberg et al., 2014** | RRID:ZFIN_ZDB-GENO-141008-1 | |
| Strain, strain background (*Danio rerio*) | Tg(brn3c:GFP) | **Xiao et al., 2005** | RRID:ZFIN_ZDB-ALT-050728-2 | |
| Strain, strain background (*Danio rerio*) | Tg(mnx1:GFP) | **Rastegar et al., 2008** | RRID:ZFIN_ZDB-GENO-140605-2 | |

*Continued on next page*

*Continued*

| Reagent type (species) or resource | Designation | Source or reference | Identifiers | Additional information |
|---|---|---|---|---|
| Strain, strain background (*Danio rerio*) | *Tg(hsp70:dnIGF1Ra-GFP)* | **Kamei et al., 2011** | RRID:ZFIN_ZDB-GENO-110614-4 | |
| Strain, strain background (*Danio rerio*) | *Tg(hsp70:pappaa-GFP)* | this paper | | Materials and methods subsection maintenance of zebrafish |
| Strain, strain background (*Danio rerio*) | *TLF* | Zebrafish International Resource Center (ZIRC) | RRID:ZFIN_ZDB-GENO-990623-2 | |
| Antibody | anti-myosinVI (rabbit polyclonal) | Proteus biosciences | RRID:AB_10013626 | 1:200 |
| Antibody | anti-SOX2 (rabbit polyclonal) | Abcam | RRID:AB_2341193 | 1:200 |
| Antibody | anti-GFP (rabbit polyclonal) | ThermoFisher Scientific | RRID:AB_221569 | 1:500 |
| Antibody | Alexa 488, secondary (rabbit polyclonal) | ThermoFisher Scientific | RRID:AB_2576217 | 1:500 |
| Peptide, recombinant protein | IGF-1 | Cell sciences | Catalog number: DU100 | |
| Other | FM1-43 | ThermoFisher Scientific | Catalog number: T3136 PubChem CID:6508724 | 3 uM for 30 s |
| Other | CellROX deep red | ThermoFisher Scientific | Catalog number: C10422 | 10 uM for 60 min |
| Other | mitoSOX red | ThermoFisher Scientific | Catalog number: M36008 | 1 uM for 30 min |
| Other | mitotracker greenFM | ThermoFisher Scientific | Catalog number: M7514 PubChem CID:70691021 | 100 nM for 5 min |
| Other | TMRE | ThermoFisher Scientific | Catalog number: T669 PubChem CID:2762682 | 25 nM for 20 min |
| Other | 0.25% trypsin-EDTA | Sigma-Aldrich | Catalog number: T3924 | |
| Other | TRIzol | Invitrogen | Catalog number: 15596026 | |
| Other | Sso fast Eva Green Supermix | BioRad | Catalog number: 1725200 | |
| Chemical compound, drug | Neomycin | Sigma-Aldrich | Catalog number: N1142 PubChem CID:24897464 | |
| Chemical compound, drug | NBI-31772 | Fisher Scientific | Catalog number: 519210 | |
| Chemical compound, drug | antimycin-a | Sigma-Aldrich | Catalog number: A8674 PubChem CID: 24891355 | |
| Chemical compound, drug | CsA | Abcam | Catalog number: ab120114 PubChem ID: 5284373 | |

*Continued on next page*

*Continued*

| Reagent type (species) or resource | Designation | Source or reference | Identifiers | Additional information |
|---|---|---|---|---|
| Chemical compound, drug | mitoTEMPO | Sigma-Aldrich | Catalog number: SML0737 PubChem CID: 134828258 | |
| Chemical compound, drug | NVP-AEW541 | Selleck | Catalog number: S1034 PubChem CID: 11476171 | |
| Commercial assay or kit | SuperScript II Reverse Transcriptase | Invitrogen | Catalog number: 18064014 | |
| Software, algorithm | Fluoview (FV10-ASW 4.2) | Olympus | RRID:SCR_014215 | |
| Software, algorithm | ImageJ | PMID: 22743772 | RRID:SCR_003070 | |
| Software, algorithm | GraphPad Prism | GraphPad | RRID:SCR_002798 | |

## Maintenance of zebrafish

To generate $pappaa^{+/+}$ and $pappaa^{p170}$ larvae for experimentation, adult $pappaa^{p170/+}$ zebrafish (on a mixed Tubingen long-fin (*TLF*), WIK background) were crossed into the following transgenic zebrafish backgrounds: $Tg(brn3c:GFP)^{s356t}$, $Tg(hsp70:dnIGF1Ra-GFP)^{zf243}$, $Tg(hsp70:pappaa-GFP),Tg(mnx1:GFP)^{ml5}$, and $Tg(myo6b:mitoGCaMP3)^{w78}$ and then incrossed. To establish the *Tg(hsp70:pappaa-GFP)* line, used gateway cloning vectors (Invitrogen) to generate a transgenesis construct under the control of a ubiquitous heat shock promoter (*hsp70*). The *pappaa* sequence was generated from cDNA of 120 hpf *TLF* embryos. Expression was assessed by fluorescence of a co-translated and post-translationally cleaved green GFP after induction by exposure in a heated water bath for 1 hr. The stable *Tg(hsp70:pappaa-GFP)* line was maintained on a $pappaa^{p170/+}$ background.. Embryonic and larval zebrafish were raised in E3 media (5 mM NaCl, 0.17 mM KCl, 0.33 mM $CaCl_2$, 0.33 mM $MgSO_4$, pH adjusted to 6.8–6.9 with $NaHCO_3$) at 29°C on a 14 hr/10 hr light/dark cycle through 5 days post fertilization (dpf) (*Kimmel et al., 1995*; *Gyda et al., 2012*). All experiments were done on larvae between 4–6 dpf. Genotyping of $pappaa^{p170}$ larvae was performed as previously described (*Wolman et al., 2015*).

## Pharmacology

The following treatments were performed on *Tg(brn3c:GFP)* larvae through the addition of compounds to the larvae's E3 media at five dpf unless otherwise noted. Neomycin sulfate solution (Sigma-Aldrich N1142) was added at 1–30 μM for 1 hr. Cyclosporin A (Abcam ab120114; dissolved in DMSO) was added at 0.3–3 μM for 1 hr. Antimycin A (Sigma-Aldrich A8674; dissolved in DMSO) was added at 100–500 pM for 24 hr, beginning at four dpf. MitoTEMPO (Sigma-Aldrich SML0737; dissolved in DMSO) was added at 10–100 μM 30 min prior to a 1 hr exposure to 10 μM neomycin. To stimulate IGF1 signaling: larvae were pre-treated with either NBI-31772 at 30–100 μM (Fisher Scientific 519210; dissolved in DMSO), or recombinant IGF1 at 10–100 ng/mL (Cell Sciences DU100; dissolved in 10 μM HCl and diluted in 0.1 mg BSA in E3) for 24 hr prior (beginning at four dpf) and then exposed to 10 μM neomycin for 1 hr on five dpf. To inhibit IGF1 signaling, larvae were treated with NVP-AEW541 at 30 μM (Selleck S1034; dissolved in DMSO) for 24 hr (beginning at four dpf), before live imaging. Following each treatment period, larvae were washed 3 times with E3 and left to recover in E3 for 4 hr at 28°C before fixation with 4% paraformaldehyde (diluted to 4% w/v in PBS from 16% w/v in 0.1M phosphate buffer, pH 7.4). For mitoTEMPO, NBI-31772, and IGF1 treatments, the compounds were re-added to the E3 media for the 4 hr recovery period post neomycin washout. Vehicle-treated controls were exposed to either 0.9% sodium chloride in E3 (neomycin control), 0.1 mg BSA in E3 (IGF1 control) or 0.1% DMSO in E3 for the other compounds.

## Hair cell survival

Hair cell survival experiments were performed in *Tg(brn3c:GFP))*, *TLF*, *Tg(hsp70:dnIGF1R-GFP)*, *or Tg(hsp70:pappaa-GFP)* larvae where hair cells are marked by GFP (*brn3c*) or anti-myosin VI antibody (*TLF*, *hsp70:dnIGF1Ra-GFP*, and *hsp70:pappaa-GFP*). For each larva, hair cells were counted from the same three stereotypically positioned neuromasts (IO3, M2, and OP1) (*Raible and Kruse, 2000*) and averaged. The percent of surviving hair cells was calculated as: [(mean number of hair cells after treatment)/ (mean number of hair cells in vehicle treated group)] X 100. For the heat shock experiments, treatment groups were normalized to the non-heat-shocked, vehicle-treated group for each genotype. Normalizations were genotype specific to account for a slight increase in hair cell number (~2 per neuromast) in *pappaa$^{p170}$* larvae at five dpf.

## Induction of transgenic dominant negative IGF1Ra and Pappaa expression

To induce expression of a dominant negative form of IGF1Ra, we used *Tg(hsp70:dnIGF1Ra-GFP)* larvae, which express *dnIGF1Ra-GFP* under the control of zebrafish *hsp70* promoter (*Kamei et al., 2011*). *dnIGF1Ra-GFP* expression was induced by a 1 hr heat shock at 37°C, which was performed once per 12 hr from either 24 hpf to five dpf or from 4 dpf to five dpf. To induce expression of Pappaa, *Tg(hsp70:pappaa-GFP)* larvae in the *pappaa$^{p170}$* background were heat shocked once for 30 min at 37°C at four dpf. To control for possible effects of heat shocking, non-transgenic wild type and *pappaa$^{p170}$* larvae were exposed to the same treatment.

## Single cell dissociation and fluorescence activated cell sorting

For each genotype, 30 five dpf *Tg(mnx1:GFP)* and 200 five dpf *Tg(brn3c:GFP)* larvae were rinsed for 15 min in Ringer's solution (116 mM NaCl, 2.9 mM KCl, 1.8 mM CaCl$_2$, 5 mM HEPES, pH 7.2) (*Guille, 1999*). *pappaa$^{p170}$* larvae were identified by lack of swim bladder inflation (*Wolman et al., 2015*). To collect motor neurons we used whole *Tg(mnx1:GFP)* larvae and to collect hair cells we used tails dissected from *Tg(brn3c:GFP)* larvae. Samples were pooled into 1.5 mL tubes containing Ringer's solution on ice, which was then replaced with 1.3 mL of 0.25% trypsin-EDTA (Sigma-Aldrich) for digestion. *Tg(mnx1:GFP)* samples were incubated for 90 min and *Tg(brn3c:GFP)* were incubated for 20 min. Samples were titrated gently by P1000 pipette tip every 15 min for motor neurons and every 5 min for hair cells. To stop cell digestion, 200 μL of 30% FBS and 6 mM CaCl$_2$ in PBS solution (*Steiner et al., 2014*) was added, cells were centrifuged at 400 g for 5 min at 4°C, the supernatant was removed, the cell pellet was rinsed with Ca$^{2+}$-free Ringer's solution and centrifuged again. The cell pellet was then resuspended in 1X Ca$^{2+}$-free Ringer's solution (116 mM NaCl, 2.9 mM KCl, 5 mM HEPES, pH 7.2) and kept on ice until sorting. Immediately before sorting, cells were filtered through a 40 μm cell strainer and stained with DAPI. A two-gates sorting strategy was employed. DAPI was used to isolate live cells, followed by a forward scatter (FSC) and GFP gate to isolate GFP+ cells. Sorted cells were collected into RNAse-free tubes containing 500 μL of TRIzol reagent (Invitrogen) for RNA extraction.

## RNA extraction and RT-PCR

Total RNA was extracted from whole larvae and FACS sorted motor neurons and hair cells using TRIzol. cDNA was synthesized using SuperScript II Reverse Transcriptase (Invitrogen 18064014). Real-time Quantitative PCR (RT-qPCR) was performed using Sso fast Eva Green Supermix (Biorad 1725200) in a StepOnePlus Real-Time PCR System (Applied Biosystems) based on manufacture recommendation. Reactions were run in 3–4 technical replicates containing cDNA from 50 ng of total RNA/reaction. The primer sequences for the antioxidant genes were previously described (*Jin et al., 2010*) and are as follows: For *sod1*, forward: GTCGTCTGGCTTGTGGAGTG and reverse: TGTCAGCGGGCTAGTGCTT; for *sod2*, forward: CCGGACTATGTTAAGGCCATCT and reverse: ACACTCGGTTGCTCTCTTTTCTCT; for *gpx*, forward: AGATGTCATTCCTGCACACG and reverse: AAGGAGAAGCTTCCTCAGCC; for *catalase*, forward: AGGGCAACTGGGATCTTACA and reverse: TTTATGGGACCAGACCTTGG. *b-actin* was used as an endogenous control with the following primer sequences: forward TACAGCTTCACCACCACAGC and reverse: AAGGAAGGCTGGAAGAGAGC (*Wang et al., 2005*). Cycling conditions were as follows: 1 min at 95°C, then 40 cycles of 15 s at 95°C, followed by 1 min at 60°C (*Jin et al., 2010*): Relative quantification of gene expression was done

using the $2^{-\Delta\Delta Ct}$ method (*Livak and Schmittgen, 2001*). PCR amplification for *pappaa* fragment was performed by using forward primer: AGACAGGGATGTGGAGTACG, and reverse primer: GTTGCA-GACGACAGTACAGC. PCR conditions were as follows: 3 min at 94℃, followed by 40 cycles of 94℃ for 30 s, 57℃ for 1 min, and 70℃ for 1 min (*Wolman et al., 2015*). The PCR product was run on a 3% agarose gel.

## Live imaging

All experiments were done on 5–6 dpf *pappaa*$^{p170}$ and *pappaa*$^{+/+}$ larvae at room temperature. Images were acquired with an Olympus Fluoview confocal laser scanning microscope (FV1000) using Fluoview software (FV10-ASW 4.2). To detect oxidative stress, *Tg(brn3c:GFP)* larvae were incubated in 10 µM CellROX Deep Red (Thermofischer Scientific C10422; dissolved in DMSO) and 1 µM mito-SOX Red (Thermofischer Scientific M36008; dissolved in DMSO) in E3 for 60 min and 30 min, respectively. To detect mitochondrial transmembrane potential, *Tg(brn3c:GFP)* larvae were incubated in 25 nM TMRE (Thermofischer Scientific T669; dissolved in DMSO) for 20 min. To investigate the effects of inhibiting IGF1 signaling on mitochondrial transmembrane potential, larvae were incubated in 25 nM TMRE for 20 min following treatment with NVP-AEW541. To detect MET channel function, *Tg(brn3c:GFP)* larvae were incubated in 3 µM FM1-43 (Thermofischer Scientific T3136; dissolved in DMSO) for 30 s. To measure mitochondrial mass, larvae were incubated in 100 nM mitotracker green FM (Thermofischer Scientific M7514; dissolved in DMSO) for 5 min. Following the incubation period, larvae were washed three times in E3, anesthetized in 0.002% tricaine (Sigma-Aldrich) in E3, and mounted as previously described (*Stawicki et al., 2014*). Fluorescent intensity of the reporter was measured using ImageJ (*Schneider et al., 2012*) by drawing a region of interest around *brn3c:GFP*-labaled hair cells of the neuromast from Z-stack summation projections that included the full depth of the hair cells. Background fluorescent intensity was measured by drawing a ROI away from the neuromast in the same Z-stack summation projection. The corrected total cell fluorescence (CTCF) was used to subtract background fluorescence from each reporter. The CTCF formula was as follows: Integrated Density - (Area of selected cells X Mean fluorescence of background) (*McCloy et al., 2014*). The mean CTCF of each live dye was reported both independently and as the ratio to the mean CTCF of GFP fluorescence.

## Immunohistochemistry and in situ hybridization

For whole-mount immunostaining, larvae at five dpf were fixed in 4% paraformaldehyde for 1 hr at room temperature then rinsed three times with PBS. Larvae were blocked for 1 hr at room temperature in incubation buffer (0.2% bovine serum albumin, 2% normal goat serum, 0.8% Triton-X, 1% DMSO, in PBS, pH 7.4). Larvae were incubated in primary antibodies in IB overnight at 4℃. Primary antibodies were as follows: hair cells (anti-myosinVI, 1:200, rabbit polyclonal, Proteus biosciences, RRID:AB_10013626) or using *Tg(brn3c:GFP)* larvae (anti-GFP, 1:500, rabbit polyclonal; ThermoFisher Scientific, RRID:AB_221569), and support cells (anti-SOX2 ab97959, 1:200, rabbit polyclonal; Abcam, RRID:AB_2341193) (*He et al., 2014*). Following incubation of primary antibodies, larvae were incubated in AlexaFluor-488- conjugated secondary antibody in IB for 4 hr at room temperature. (goat anti-rabbit polyclonal, 1:500; ThermoFisher Scientific, RRID:AB_2576217). After staining, larvae were mounted in 70% glycerol in PBS. Images were acquired with an Olympus Fluoview confocal laser scanning microscope (FV1000) using Fluoview software (FV10-ASW 4.2).

For whole-mount in situ hybridization: digoxygenin-UTP-labeled antisense riboprobes for *pappaa* (*Wolman et al., 2015*) were used as previously described (*Halloran et al., 1999*; *Chalasani et al., 2007*). Images of colorimetric in situ reactions were acquired using a Leica Fluorescence stereo microscope with a Leica DFC310 FX digital color camera. Images of fluorescent in situ reactions were acquired using an Olympus Fluoview confocal laser scanning microscope (FV1000).

## Statistics

All data were analyzed using GraphPad Prism Software 7.0b (GraphPad Software Incorporated, La Jolla, Ca, USA, RRID:SCR_002798). Prior to use of parametric statistics, the assumption of normality was tested using Shapiro-Wilk's test. Parametric analyses were performed using a two-tailed unpaired *t*-test with Welch's correction, multiple *t* tests with a Holm-Sidak correction, or 2-way ANOVA with a Holm-Sidak correction. Data are presented as means ± standard error of the mean

(SEM; N = sample size). Significance was set at $p < 0.05$. N for each experiment is detailed in the figure legends. All data presented are from individual experiments except for data in *Figures 1B*, *3B* and *6E*. Data collected from multiple experiments were normalized to their respective controls prior to pooling.

## Acknowledgements

The authors would like to thank Dr. David Raible (University of Washington-Seattle) for the *myo6b: mitoGCaMP3* fish line and Dr. Corinna Burger (University of Wisconsin Department of Neurology) for use of the RT-qPCR cycler.

## Additional information

### Funding

| Funder | Author |
| --- | --- |
| Greater Milwaukee Foundation | Marc A Wolman |
| Saudi Ministry of Education | Mroj Alassaf |

The funders had no role in study design, data collection and interpretation, or the decision to submit the work for publication.

### Author contributions

Mroj Alassaf, Conceptualization, Data curation, Formal analysis, Investigation, Writing—original draft; Emily C Daykin, Data curation; Jaffna Mathiaparanam, Resources, Methodology; Marc A Wolman, Conceptualization, Resources, Funding acquisition, Project administration, Writing—review and editing

### Author ORCIDs

Marc A Wolman https://orcid.org/0000-0002-8929-779X

### Ethics

Animal experimentation: This study was performed in strict accordance with the recommendations in the Guide for the Care and Use of Laboratory Animals of the National Institutes of Health. All of the animals were handled according to approved institutional animal care and use committee (IACUC) protocols (L00457-A1) of the University of Wisconsin.

### Decision letter and Author response

Decision letter https://doi.org/10.7554/eLife.47061.032
Author response https://doi.org/10.7554/eLife.47061.033

## Additional files

### Supplementary files

• Transparent reporting form
DOI: https://doi.org/10.7554/eLife.47061.030

### Data availability

All data generated or analysed during this study are included in the manuscript and supporting files. Source data files have been provided for Figures 1,3,4,5,6,7 and all supplementary figures.

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
