## [Decision Letter]

[Editors’ note: a previous version of this study was rejected after peer review, but the authors submitted for reconsideration. The first decision letter after peer review is shown below.]

Thank you for submitting your work entitled "Pregnancy-associated plasma protein-aa promotes neuron survival by regulating mitochondrial function" for consideration by *eLife*. Your article has been reviewed by a Senior Editor, a Reviewing Editor, and three reviewers. The following individuals involved in review of your submission have agreed to reveal their identity: David Raible (Reviewer #1).

Our decision has been reached after consultation between the reviewers. Based on these discussions and the individual reviews below, we regret to inform you that your work will not be considered for publication in *eLife* at this time. While there was general enthusiasm for the studies, there were a number of concerns raised by the reviewers that we feel would not be able to be addressed in the standard period given for revision. However, given the overall enthusiasm we would consider a revised version of the manuscript if you can meet the referees' concerns. Please note that this would be considered a new submission.

*Reviewer #1:*

Alassaf et al., examine the effects of mutation in the IGF signaling regulator Pappaa on hair cell survival after exposure to neomycin. They find that *pappaa* mutants are more sensitive to damage. They present evidence that hair cells in mutants have higher levels of ROS production, increases in mitochondrial polarization and increases in mitochondrial calcium, all previously implicated in neomycin-induced hair cell death. They also present evidence that manipulating IGF signaling alters the effects of the *pappaa* mutation on hair cell survival. Finally, they show that *pappaa* mutants lose motor neurons over time. Several issues need to be addressed before the manuscript could be considered for publication.

The authors propose that IGF signaling is important for hair cell survival but do not test this directly. They have used the dn-igfr transgenic line in previous publications – does this method of inhibiting IGF signaling also alter hair cell survival? Does the dn-igfr or the IGF antagonist NVP-AEW541 shift the neomycin dose-response curve? Does SC79 and IGF-1 protect wildtype hair cells against neomycin?

Other points:

The authors should refer to hair cells in the abstract rather than describing them as neurons.

The authors suggest that the failure to add new hair cells is because of degeneration (Figure 1). However, growth of the lateral line depends on the ability of larvae to feed. Are mutants able to feed and grow normally? If not, this phenotype is likely indirect.

The analysis to determine whether Pappaa is expressed in hair cells (Figure 2) is inconclusive. Expression centrally in neuromasts (2A) usually corresponds to hair cells. Co-localization with GFP is inconclusive (2B). There is no direct comparison between hair cells and support cells by RT-PCR. Comparison to whole larva is not relevant.

Experiments shown in 3G are inconclusive. Previous work has shown that mitoTEMPO protects hair cells from neomycin exposure in wildtype animals. Are *pappaa* mutants differentially sensitive to this treatment?

Additional experiments are needed for the analysis of IGFR signaling (Figure 5A). The specificity of the phospho-IGF1R staining should be tested. This staining should be sensitive to NVP-AEW541. It is also important to determine whether staining is actually in hair cells.

Controls are missing from pharmacological experiments (Figure 5). The effect of NVP-AEW541 on hair cells in the absence of neomycin should be shown (5D), and conditions repeated on *pappaa* mutants. The effect of SC79 (5E) and IGF-1 (5F) on wildtype animals should be shown as discussed above. In particular the authors should test whether these reagents are differentially effective in wildtype and *pappaa* mutant embryos.

*Reviewer #2:*

The topic of this study is of great interest to both auditory neuroscientists and to the broader neuroscience community. With that said, hair cells, while they share properties with neurons, are highly specialized and have their own distinct properties. As the data on mitochondrial function and the contribution of IGF1R signaling to survival were collected from hair cells, the emphasis on "neurons" in place of "hair cells" in the title and the manuscript is misleading. Additionally, I have major concerns about how the authors performed and analyzed the live imaging experiments.

Point by point concerns are addressed below:

- Naturally occurring cell death (Figure 1): Couldn't this observation also indicate a failure to proliferate?

- FM1-43 uptake (Figure 3 A-B): The dye was diluted with DMSO, which confounds the results as DMSO could be facilitating entry into hair cells.

- Image analysis (Figure 3 B,D,F; Figure 4 C,D; Figure 5C): The method used for analyzing differences in the fluorescence of indicators relative to brn3c:GFP fluorescence (F indicator/ FGFP) is unconventional. I'm concerned that using GFP fluorescence to normalize for variable neuromast size would give misleading results; variances in hair-cell GFP fluorescence appears completely independent of variances in indicator fluorescence (see Figure 3E for an example). Is this a technique that has been previously used? If not, the authors need to clearly explain their rationale. Additionally, the authors need to provide more detail about how their images were processed and analyzed in ImageJ (e.g. were z-stack projections used? how was background fluorescence subtracted? which specific plugins were used for the analysis?)

- Subsection “Pappaa regulates mitochondrial Ca^2+^ uptake and transmembrane potential”: replace "a doubling" with " ~2-fold greater".

- Subsection “Pappaa-IGF1 receptor signaling is required for neuron survival”: "based on their numbers" – the observation that there are more hair cells in *pappaa^p170^* mutants relative to their siblings could indicate that the neuromasts are not developing normally.

- Subsection “Pappaa-IGF1 receptor signaling is required for neuron survival”: "regulation of mitochondrial function in neurons" – the previous paragraph in the discussion suggests, since Pappaa is expressed in spinal motor neurons but not hair cells, that Pappaa may promote motor neuron survival via mechanisms that differ from hair cells. How do the authors extrapolate that the evidence they show for mitochondrial dysfunction contributing to hair-cell loss in the mutant also applies to loss of neurons?

- Pharmacology: 1% DMSO seems unusually high (usually 5 to 10x less is used as a carrier for free-swimming larvae). Why did the authors choose that concentration?

*Reviewer #3:*

The manuscript "Pregnancy-associated plasma protein-aa promotes neuron survival by regulating mitochondrial function" is a thorough assessment of both hair cell and neuron presence and measures of mitochondrial health in vivo with loss of the extracellular protein Pappaa using the zebrafish system. Using in vivo imaging of mitochondrial/cytoplasmic ROS, mitochondrial calcium, and mitochondrial membrane potential, the authors demonstrate that loss of Pappaa leads to alterations in these measures of mitochondrial health. These abnormal mitochondrial measures correlate with loss of neurons and hair cells in older larvae as well as increased susceptibility of hair cells to calcium and neomycin-mediated death. Strikingly, ROS reduction decreases hair cell loss in *pappaa* mutants, confirming the relationship between mitochondrial ROS and hair cell loss. Overall, this is a thorough description of a novel role for the extracellular factor Pappaa in hair cell and neuron presence. The manuscript is well written and flows nicely. The data presented are solid and offer potential new insights into the role of IGF in maintenance of neurons and hair cells in vivo. However, a few additional experiments would solidify this role, specifically making the connection between cell survival and *pappaa* mutation and addressing the known role of IGF in mitochondrial biology. Suggestions for modifications to strengthen this work to make it suitable for publication are below.

Essential revisions:

1) The authors argue that loss of Pappaa function causes decreased numbers of hair cell and neurons; however, while number of hair cell and motor neurons is decreased by 9 dpf, no measures of cell death in the absence of chemical ablation are shown. Additionally, no measures of proliferation are shown. To clarify the basic function of Pappaa in neuron presence, a measure of proliferation for hair cells and motor neurons as well as a measure of cell death in these populations should be shown. Numerous methods exist to assay these cellular processes in vivo in zebrafish larvae.

2) IGF signaling has been implicated in the regulation of mitochondrial dynamics, mitophagy and mitochondrial biogenesis. Disrupting these processes could lead to mitochondrial stress, which would then lead to the calcium level, ROS, and matrix potential defects observed in *pappaa* mutants. Essentially sensitizing the system. To determine if defects in these aspects of mitochondrial biology are the primary drivers of the hair cell and neuron loss, the authors should look at mitochondrial size, mitochondrial load, and mitochondrial localization using either a transient transgenic approach or mitotracker.

3) The majority of the work presented was done in hair cells and the results are then assumed to represent what is also happening in motor neurons. To determine if this is indeed the case, at least a subset of the measures of mitochondrial calcium, potential, and ROS should be done in motor neurons of *pappaa* mutants. Additionally, the ability of exogenous IGF or AKT to rescue motor neurons should be shown to confirm or refute their similarity to hair cells.

4) In subsection “Pappaa-IGF1 receptor signaling is required for neuron survival” the authors argue that hair cells and motor neurons develop normally to day 5, indicating a role for Pappaa in maintenance rather than development; however, it is not shown if Pappaa is maternally expressed and, if it is, how long the protein lasts. This would be essential for this argument as it is entirely possible that maternal Pappaa allows for normal development rather than Pappaa not being necessary for development. Alternatively, a maternal zygotic mutant could also answer this question.

[Editors’ note: what now follows is the decision letter after the authors submitted for further consideration.]

Thank you for submitting your article "Pregnancy-associated plasma protein-aa supports hair cell survival by regulating mitochondrial function" for consideration by *eLife*. Your article has been reviewed by Didier Stainier as the Senior Editor, a Reviewing Editor, and two reviewers. The reviewers have opted to remain anonymous.

The reviewers have discussed the reviews with one another and the Reviewing Editor has drafted this decision to help you prepare a revised submission.

Summary:

The revised manuscript "Pregnancy-associated plasma protein-aa supports hair cell survival by regulating mitochondrial function" carefully describes the function of IGF signaling in hair cell resistance to cell death. The authors have convincingly shown that loss of Pappaa results in increased susceptibility to neomycin-mediated hair cell death and that this is likely through increased susceptibility to mitochondrial stress pathways. Additionally, the authors have strengthened the work by showing that Pappaa functions in the IGF availability pathway through using the IGFBP inhibitor NBI-31772. Together, their work argues for a role for IGF signaling in the maintenance of hair cell survival through supporting mitochondria/inhibiting mitochondrial ROS stress.

Essential revisions:

1) The reviewers again raised concerns about the numbers of replicates, with no clear way to know how many times experiments were performed. In addition, experiments appear to be often underpowered with small n's. These are of particular concern for imaging experiments presented in Figure 5 and Figure 6. The authors need to clearly state the numbers of samples used for calculations, the numbers of experiments performed, and provide justification.

2) In a previous version of this manuscript, exogenous IGF was shown to suppress neomycin-mediated hair cell death. It is unclear why this was taken out as it clearly compliments Figure 3C which shows that enhancing IGF signaling can suppress hair cell death. It should at least be included in the supplement unless there is a specific reason to exclude it.

3) Throughout the manuscript, the authors use "% surviving cells" as a measure of hair cell survival. This implies that they counted the hair cells in the same neuromasts before and after treatment. Their methods state that they normalized number of cells after treatment to average number of cells in the vehicle treated control. While this is a suitable way to control for the data, this should be spelled out in the Results section rather than hidden in the Materials and methods section.

1) Subsection “Pappaa regulates survival of hair cell sensory neurons”, the authors state: "This live probe (re:TMRE) emits more fluorescence as the membrane potential becomes more negative." This is not true. This vital cationic dye just accumulates to a higher degree in mitochondria that have a more negative matrix potential. The dye does not change the its emission profile to my knowledge. The language should be changed to reflect this.

2) In the revised manuscript, the authors provide "absolute fluorescence levels" and refer to these articles (Ogun and Zallocchi, 2014; Majumder et al., 2017). Both studies quantified relative FM1-43 fluorescence by performing background subtraction of fluorescence, selecting the region of interest (e.g. neuromast), and measuring the average fluorescent intensity i.e. total fluorescent intensity/area. If this method is what the authors also used, they should update their Material and Methods section and add citations accordingly.

---

## [Author Response]

[Editors’ note: the author responses to the first round of peer review follow.]

Reviewer #1:[…] The authors propose that IGF signaling is important for hair cell survival but do not test this directly. They have used the dn-igfr transgenic line in previous publications – does this method of inhibiting IGF signaling also alter hair cell survival? Does the dn-igfr or the IGF antagonist NVP-AEW541 shift the neomycin dose-response curve? Does SC79 and IGF-1 protect wildtype hair cells against neomycin?

Per this suggestion, we used the *hsp70:dnIGF1Ra-GFP* transgenic line to reduce IGF1R signaling and ask whether IGF1 signaling mediates hair cell survival. As shown in Figure 1B and Figure 4C, we found that induction of dnIGF1Ra increases hair cell death in larvae exposed to neomycin and therefore shifts the neomycin dose-response curve. We have excluded the IGF-1 and SC79 supplementation experiments from the manuscript because we wanted to focus mainly on the influence of endogenous IGF1 signaling and the pathway’s extracellular regulation (Pappaa, Figure 3C), rather than exogenous supplementation by IGF1 and SC79.

The authors should refer to hair cells in the Abstract rather than describing them as neurons.

The manuscript has been edited to reflect these changes and now solely focuses on the effect of Pappaa-IGF1 signaling on hair cells.

The authors suggest that the failure to add new hair cells is because of degeneration (Figure 1). However, growth of the lateral line depends on the ability of larvae to feed. Are mutants able to feed and grow normally? If not, this phenotype is likely indirect.

We thank the reviewer for calling our attention to this important point. *pappaa* mutants do show reduced feeding. Therefore, we have excluded this result and its interpretation from the manuscript.

The analysis to determine whether pappaa is expressed in hair cells (Figure 2) is inconclusive. Expression centrally in neuromasts (2A) usually corresponds to hair cells. Co-localization with GFP is inconclusive (2B). There is no direct comparison between hair cells and support cells by RT-PCR. Comparison to whole larva is not relevant.

We have modified Figure 2 to provide more conclusive expression data. The colorimetric *pappaa* in situ expression shows clear expression by the neuromasts. We believe the expression centrally in each neuromast is by the inner support cells, not the hair cells. By analyzing Z-stack confocal images slice by slice in *Brn3c-GFP* larvae stained for *pappaa* expression by FISH, we do not observe *pappaa mRNA* in hair cells. We performed the RT-PCR of *pappaa* on FAC sorted hair cells to rule out the possibility that hair cells express *pappaa*, but at a level below FISH detection. RT-PCR on whole larvae and FAC sorted motor neurons provided positive controls. Unfortunately, we do not have a transgenic line that only labels support cells to use in a complementary experiment.

Experiments shown in 3G are inconclusive. Previous work has shown that mitoTEMPO protects hair cells from neomycin exposure in wildtype animals. Are pappaa mutants differentially sensitive to this treatment?

Results of the mitoTEMPO experiment are now in Figure 7B. Consistent with previous work (Esterberg et al., 2016), we find mitoTEMPO protects hair cells from neomycin exposure in wildtype and *pappaa* mutants. It is difficult to determine relative sensitivity of the two genotypes to mitoTEMPO because mitoTEMPO produces an all or nothing survival effect in both genotypes at doses ranging from 10-100 μM. Rather than assessing relative sensitivity, the rationale behind our experiment was to determine whether the excessive ROS production by *pappaa* mutant hair cells was the root cause of their increased death when exposed to neomycin. If mitoTEMPO did not or only partially rescued hair cell loss in *pappaa* mutants, then this outcome would have suggested the mutant hair cells have additional defects that underlie their increased neomycin sensitivity. Given that mitoTEMPO provided up to complete protection against neomycin, we concluded that ROS was the primary reason.

Additional experiments are needed for the analysis of IGFR signaling (Figure 5A). The specificity of the phospho-IGF1R staining should be tested. This staining should be sensitive to NVP-AEW541. It is also important to determine whether staining is actually in hair cells.

The anti-pIGF1R antibody used was validated by Chablais and Jazwinska (2010) in zebrafish using NVP-AEW541. However, we decided to exclude the result (reduced p-IGF1R labeling in *pappaa* mutant hair cells) because it does not significantly add to the manuscript.

Controls are missing from pharmacological experiments (Figure 5). The effect of NVP-AEW541 on hair cells in the absence of neomycin should be shown (5D), and conditions repeated on pappaa mutants. The effect of SC79 (5E) and IGF-1 (5F) on wildtype animals should be shown as discussed above. In particular the authors should test whether these reagents are differentially effective in wildtype and pappaa mutant embryos.

We did not observe hair cell death when larvae were treated with NVP-AEW541 alone. That said, these controls and experiments are now moot, since we have decided against including the results in the manuscript. As detailed above, we instead used the *hsp70:dnIGF1Ra-GFP* transgenic line to reduce IGF1R signaling and ask whether IGF1 signaling mediates hair cell survival (Figure 1B and Figure 4C). Notably, expression of *dnIGF1Ra* alone (no neomycin) did not affect hair cell survival. Because we decided to focus more on the extracellular regulation of endogenous IGF1 signaling rather than exogenous supplementation of IGF1 and SC79, we included results from pharmacological stimulation of IGF1 bioavailability by NBI-31772 (Figure 3C).

Reviewer #2:[…] Point by point concerns are addressed below:- Naturally occurring cell death (Figure 1): Couldn't this observation also indicate a failure to proliferate?

The hair cells in *pappaa* mutants develop on time and are capable of regeneration. Therefore, we do not believe that a failure to proliferate is causing the lack of age-related increase in hair cells. However, as reviewer 1 pointed out, growth of lateral line hair cells depends on the larvae’s ability to grow and feed normally. Given that *pappaa* mutant larvae show reduced ability to feed, we have decided to exclude this result and interpretation from the current manuscript.

- FM1-43 uptake (Figure 3 A-B): The dye was diluted with DMSO, which confounds the results as DMSO could be facilitating entry into hair cells.

FM1-43 was dissolved in DMSO, as per manufacturer recommendation. We used FM1-43 at a low final concentration (3uM) made from a 5 mM stock in DMSO. The final concentration of DMSO would have been at 0.06%. This method has been used previously to demonstrate that FM1-43 (dissolved in DMSO) only enters hair cells via MET channels when exposure time is less than 1 minute (Gale et al., 2001).

- Image analysis (Figure 3 B,D,F; Figure 4 C,D; Figure 5C): The method used for analyzing differences in the fluorescence of indicators relative to brn3c:GFP fluorescence (F indicator/ FGFP) is unconventional. I'm concerned that using GFP fluorescence to normalize for variable neuromast size would give misleading results; variances in hair-cell GFP fluorescence appears completely independent of variances in indicator fluorescence (see Figure 3E for an example). Is this a technique that has been previously used? If not, the authors need to clearly explain their rationale.

To provide further analysis, we have now included the absolute fluorescence levels. This follows the field standard (Ogun and Zallocchi, 2014; Majumder et al., 2017). Additionally, we have included the normalized fluorescence (F indicator/F GFP), which we feel is important to control for potential variability in labeling across samples.

Additionally, the authors need to provide more detail about how their images were processed and analyzed in ImageJ (e.g. were z-stack projections used? how was background fluorescence subtracted? which specific plugins were used for the analysis?)

We have provided more detail on the image processing in subsection “Live imaging”. We have added the text “Fluorescent intensity of the reporter was measured using ImageJ (Schneider et al., 2012) by drawing a region of interest around brn3c:GFP-labaled hair cell cluster of the neuromast from Z-stack summation projections that included the full depth of the hair cells. Background fluorescent intensity was measured by drawing a ROI away from the neuromast in the same Z-stack summation projection. The corrected total cell fluorescence (CTCF) was used to subtract background fluorescence from each reporter. The CTCF formula was as follows: Integrated Density – (Area of selected cells X Mean fluorescence of background) (McCloy et al., 2014). Relative fluorescent intensity was reported as the ratio to GFP fluorescence.” No specific plugins were used.

- Subsection “Pappaa regulates mitochondrial Ca^2+^ uptake and transmembrane potential”: replace "a doubling" with " ~2-fold greater".

This change has been made.

- Subsection “Pappaa-IGF1 receptor signaling is required for neuron survival”: "based on their numbers" – the observation that there are more hair cells in pappaa^p170^ mutants relative to their siblings could indicate that the neuromasts are not developing normally.

We do not believe *pappaa* affects neuromast development because (1) the hair cell and support cell morphologies are normal (Figure 4A and Figure 3—figure supplement 1B), (2) the mutants respond to acoustic stimuli and do not show hallmark behavioral signs of dysfunctional hair cells (Wolman et al., 2015) and (3) we show that Pappaa is dispensable for hair cell development in the context of hair cell sensitivity to neomycin (Figure 3A-B). The increased number of hair cells in neuromasts of *pappaa* mutants is very modest: ~2 per neuromast. If we did not account for this difference and used absolute hair cell counts to reflect mortality, then the hair cell loss would appear artificially inflated in the *pappaa* mutants compared to wild type controls. Therefore, we normalized hair cell loss to vehicle treated control wild type or *pappaa* mutants to most accurately represent and compare the relative hair cell loss between the two genotypes.

- Subsection “Pappaa-IGF1 receptor signaling is required for neuron survival”: "regulation of mitochondrial function in neurons" – the previous paragraph in the discussion suggests, since Pappaa is expressed in spinal motor neurons but not hair cells, that Pappaa may promote motor neuron survival via mechanisms that differ from hair cells. How do the authors extrapolate that the evidence they show for mitochondrial dysfunction contributing to hair-cell loss in the mutant also applies to loss of neurons?

The current manuscript no longer includes results or discussion of Pappaa’s influence on motor neuron survival. Therefore, this section has been omitted.

- Pharmacology: 1% DMSO seems unusually high (usually 5 to 10x less is used as a carrier for free-swimming larvae). Why did the authors choose that concentration?

“1% DMSO” was a typo. Changes were made to reflect the real concentration (0.1%).

Reviewer #3:[…] Essential revisions:

*1) The authors argue that loss of Pappaa function causes decreased numbers of hair cell and neurons; however, while number of hair cell and motor neurons is decreased by 9 dpf, no measures of cell death in the absence of chemical ablation are shown. Additionally, no measures of proliferation are shown. To clarify the basic function of Pappaa in neuron presence, a measure of proliferation for hair cells and motor neurons as well as a measure of cell death in these populations should be shown. Numerous methods exist to assay these cellular processes* in vivo *in zebrafish larvae.*

Because hair cells in *pappaa* mutants develop on time and are capable of regeneration, we do not believe that failure to proliferate is causing the lack of age-related increase in hair cells. However, as reviewer 1 has pointed out, growth of lateral line hair cells depends on the larvae’s ability to grow and feed normally. Given that *pappaa* mutant larvae show reduced ability to feed, we have decided to exclude this experiment from our manuscript. We also decided to focus solely on hair cells. Results describing motor neuron degeneration have been excluded from the revised manuscript.

2) IGF signaling has been implicated in the regulation of mitochondrial dynamics, mitophagy and mitochondrial biogenesis. Disrupting these processes could lead to mitochondrial stress which would then lead to the calcium level, ROS, and matrix potential defects observed in pappaa mutants. Essentially sensitizing the system. To determine if defects in these aspects of mitochondrial biology are the primary drivers of the hair cell and neuron loss, the authors should look at mitochondrial size, mitochondrial load, and mitochondrial localization using either a transient transgenic approach or mitotracker.

We analyzed mitochondrial load using mitotracker and found no significant difference (Figure 5—figure supplement 1). A future direction of this project is to evaluate mitochondrial size, morphology, and localization in *pappaa* mutants by electron microscopy. We believe results from these analyses are beyond the scope of the current study.

3) The majority of the work presented was done in hair cells and the results are then assumed to represent what is also happening in motor neurons. To determine if this is indeed the case, at least a subset of the measures of mitochondrial calcium, potential, and ROS should be done in motor neurons of pappaa mutants. Additionally, the ability of exogenous IGF or AKT to rescue motor neurons should be shown to confirm or refute their similarity to hair cells.

In the manuscript’s revised form, we only show results for hair cells and we have limited our conclusions to hair cells.

4) In subsection “Pappaa-IGF1 receptor signaling is required for neuron survival” the authors argue that hair cells and motor neurons develop normally to day 5, indicating a role for pappaa in maintenance rather than development; however, it is not shown if pappaa is maternally expressed and, if it is, how long the protein lasts. This would be essential for this argument as it is entirely possible that maternal pappaa allows for normal development rather than pappaa not being necessary for development. Alternatively, a maternal zygotic mutant could also answer this question.

In our revised manuscript, we have included an experiment in which we induce Pappaa expression at 4 dpf (Figure 4B). We found that post-developmental expression of Pappaa is sufficient to rescue *pappaa* mutant hair cell sensitivity to neomycin.

[Editors' note: the author responses to the re-review follow.]

[…] Essential revisions:1) The reviewers again raised concerns about the numbers of replicates, with no clear way to know how many times experiments were performed. In addition, experiments appear to be often underpowered with small n's. These are of particular concern for imaging experiments presented in Figure 5 and Figure 6. The authors need to clearly state the numbers of samples used for calculations, the numbers of experiments performed, and provide justification.

We have added more details concerning the specific number of neuromasts and animals for each treatment group in the figures and in the figure legends. Please note that the N in this paper refers to the number of animals per group (shown at base of bars) and not neuromasts. Multiple neuromasts per animal were analyzed and averaged. To address the number of experimental trials, we’ve added the following in the methods “All data presented are from individual experiments except for data in Figure 1B, Figure 3B, and Figure 6E. Data collected from multiple experiments were normalized to their respective controls prior to pooling”. We also included the number of experiments in the relevant figure legends.

Because each live imaging experiment was done on the same day to control for any technical variabilities, we were limited in the number of animals tested. Please note that multiple neuromasts were analyzed per animal resulting in an average of 20 neuromasts per genotype (or treatment group). However, we do agree that some experiments had a particularly small number of animals tested (4 or fewer). Therefore, we have repeated experiments presented in Figure 5—figure supplement 1 and Figure 6B and 6D-D’ to include a bigger sample size.

2) In a previous version of this manuscript, exogenous IGF was shown to suppress neomycin-mediated hair cell death. It is unclear why this was taken out as it clearly compliments Figure 3C which shows that enhancing IGF signaling can suppress hair cell death. It should at least be included in the supplement unless there is a specific reason to exclude it.

Thank you for the suggestion. We’ve incorporated the exogenous IGF1 experiment to the revised manuscript (now figure 3C).

3) Throughout the manuscript, the authors use "% surviving cells" as a measure of hair cell survival. This implies that they counted the hair cells in the same neuromasts before and after treatment. Their methods state that they normalized number of cells after treatment to average number of cells in the vehicle treated control. While this is a suitable way to control for the data, this should be spelled out in the Results section rather than hidden in the Materials and methods section.

We have changed the Y-axis titles of relevant graphs to better clarify this.